# Transarterial Radioembolization of Hepatocellular Carcinoma, Liver-Dominant Hepatic Colorectal Cancer Metastases, and Cholangiocarcinoma Using Yttrium90 Microspheres: Eight-Year Single-Center Real-Life Experience

**DOI:** 10.3390/diagnostics11010122

**Published:** 2021-01-14

**Authors:** Julie Pellegrinelli, Olivier Chevallier, Sylvain Manfredi, Inna Dygai-Cochet, Claire Tabouret-Viaud, Guillaume Nodari, François Ghiringhelli, Jean-Marc Riedinger, Romain Popoff, Jean-Marc Vrigneaud, Alexandre Cochet, Serge Aho, Marianne Latournerie, Romaric Loffroy

**Affiliations:** 1Department of Vascular and Interventional Radiology, Image-Guided Therapy Center, François-Mitterrand University Hospital, 14 Rue Paul Gaffarel, BP 77908, 21079 Dijon, France; julie.pellegrinelli@chu-dijon.fr (J.P.); olivier.chevallier@chu-dijon.fr (O.C.); 2Department of Gastroenterology and Hepatology, François-Mitterrand University Hospital, 14 Rue Paul Gaffarel, BP 77908, 21079 Dijon, France; sylvain.manfredi@chu-dijon.fr (S.M.); marianne.latournerie@chu-dijon.fr (M.L.); 3Department of Nuclear Medicine, Georges François Leclerc Center, 1 Rue Professeur Marion, 21000 Dijon, France; idygaicochet@cgfl.fr (I.D.-C.); ctabouret@cgfl.fr (C.T.-V.); gnodari@cgfl.fr (G.N.); jmriedinger@cgfl.fr (J.-M.R.); rpopoff@cgfl.fr (R.P.); jmvrigneaud@cgfl.fr (J.-M.V.); alexandre.cochet@u-bourgogne.fr (A.C.); 4Department of Medical Oncology, Georges François Leclerc Center, 1 Rue Professeur Marion, 21000 Dijon, France; fghiringhelli@cgfl.fr; 5Department of Epidemiology and Biostatistics, François-Mitterrand University Hospital, 14 Rue Paul Gaffarel, BP 77908, 21079 Dijon, France; ludwig.aho@chu-dijon.fr

**Keywords:** hepatocellular carcinoma, colorectal cancer, liver metastases, cholangiocarcinoma, radioembolization, yttrium-90, survival, adverse events, selective internal radiation therapy

## Abstract

Liver tumors are common and may be unamenable to surgery or ablative treatments. Consequently, other treatments have been devised. To assess the safety and efficacy of transarterial radioembolization (TARE) with Yttrium-90 for hepatocellular carcinoma (HCC), liver-dominant hepatic colorectal cancer metastases (mCRC), and cholangiocarcinoma (CCA), performed according to current recommendations, we conducted a single-center retrospective study in 70 patients treated with TARE (HCC, *n* = 44; mCRC, *n* = 20; CCA, *n* = 6). Safety and toxicity were assessed using the National Cancer Institute Common Terminology Criteria. Treatment response was evaluated every 3 months on imaging studies using Response Evaluation Criteria in Solid Tumors (RECIST) or mRECIST criteria. Overall survival and progression-free survival were estimated using the Kaplan-Meier method. The median delivered dose was 1.6 GBq, with SIR-Spheres^®^ or TheraSphere^®^ microspheres. TARE-related grade 3 adverse events affected 17.1% of patients. Median follow-up was 32.1 months. Median progression-free survival was 5.6 months and median overall time from TARE to death was 16.1 months and was significantly shorter in men. Progression-free survival was significantly longer in women (HR, 0.49; 95%CI, 0.26–0.90; *p* = 0.031). Risk of death or progression increased with the number of systemic chemotherapy lines. TARE can be safe and effective in patients with intermediate- or advanced-stage HCC, CCA, or mCRC refractory or intolerant to appropriate treatments.

## 1. Introduction

Liver metastases and primary liver tumors are very common. Liver metastases occur in 35–55% of patients with colorectal cancer (CRC) and are synchronous in 25% of cases [1]. CRC is the second cause of death by cancer in France, with 17,117 estimated deaths in 2018 [2]. Although chemotherapy and targeted treatments have made progress, only surgical resection or ablative procedures allow complete remission [3]. However, only 20% of patients with CRC have resectable metastases [1,3]. The median overall survival (OS) is 22 months for patients with unresectable disease [4]. Targeted nonsurgical approaches such as Yttrium-90 transarterial radioembolization (Y90 TARE) may improve survival in patients with liver-dominant hepatic colorectal cancer metastases (mCRC) who do not respond or cannot tolerate chemotherapy or targeted transarterial chemoembolization (TACE) [5].

Hepatocellular carcinoma (HCC) is the most frequent primary liver tumor, followed by cholangiocarcinoma (CCA). In western countries, most patients with HCC have underlying liver cirrhosis, a condition associated with an annual incidence of HCC of 2–7% [6]. HCC is the main cause of death in cirrhotic patients, with a median OS of 9.4 months and a 5-year survival rate of 10–15% [7]. The Barcelona Clinic Liver Cancer (BCLC) staging system can be used to determine the appropriate treatment [8]. TACE is recommended for intermediate stage HCC (BCLC stage B) with multiple localizations but no vascular or extrahepatic extension [9]. The recommended treatment for advanced HCC with extrahepatic disease or vascular invasion (BCLC C) is sorafenib [10]. Y90 TARE may be an alternative treatment option for patients with BCLC stage B or C disease and may, in some cases, allow tumor resection or liver transplantation [11]. CCA is the second most frequent primary liver tumor and carries a poor prognosis overall [12]. Its incidence has risen rapidly over the last four decades, and in France 2000 new cases are diagnosed each year [12]. Ongoing studies assessing the potential benefits of Y90 TARE for CCA treatment have produced promising results (SIRCCA trial) [13,14].

The aim of this study was to determine the safety, efficacy and outcomes of Y90 TARE in patients with intermediate or advanced stage HCC, CCA, or mCRC who did not respond or could not tolerate conventional treatments in a tertiary academic center.

## 2. Materials and Methods 

### 2.1. Study Design

We included 70 consecutive patients in a single-center retrospective study. The patients had been treated with Y90 TARE for HCC, mCRC, or CCA between October 2012 and November 2018, at the François-Mitterrand University Hospital in Dijon, France. The patients were identified and their data obtained from the hospital patient information system (DxCare). For each patient, the decision to use Y90 TARE was taken during a multidisciplinary meeting, according to national recommendations. Informed consent to the procedure was obtained from each patient before the procedure. The institution’s ethics committee andthe need for informed patient consent were waived in compliance with French legislation on retrospective studies.

### 2.2. Eligibility Criteria

The inclusion criteria applicable to patients with all tumor types were adequate hematology and liver function test results (platelets ≥ 50 × 10^9^/L, prothrombin time > 50%, and bilirubin ≤ 2.0 mg/dL), an Eastern Cooperative Oncology Group (ECOG) performance status score of 0 to 2; and feasibility of arteriography.

The specific inclusion criterion for patients with HCC was BLCC B or C stage disease (unresectable HCC without major extrahepatic disease), with or without portal thrombosis. Patients with mCRC could be included only in the absence of concurrent external beam radiotherapy. Patients with CCA were to have either unresectable unifocal or bifocal (if located in the same lobe) cholangiocarcinoma, requiring downstaging or intolerance to systemic chemotherapy. 

Exclusion criteria for all patients were significant extrahepatic disease; a life expectancy of less than 3 months; an estimated Y90 dose delivered to the lungs ≥ 30 Gy (lung shunt > 20%), determined by 99 m technetium macroaggregated albumin (99 m Tc-MAA) scintigraphy; unmanageable extrahepatic uptake, such as digestive uptake, by 99 m Tc-MAA single-photon emission computed tomography/computed tomography (SPECT/CT); previous external beam radiation therapy to the liver; ascites or liver failure; and pregnancy.

### 2.3. Data Collection

All data were collected retrospectively. For each patient, the following data were abstracted from the patient’s gastroenterologist and/or oncologist files in the hospital database (DxCare): age, sex, histological diagnosis, ECOG score, whether extrahepatic disease was present, number and distribution of tumors, whether the patient had portal thrombosis and/or cirrhosis, the Child Pugh score in patients with cirrhosis, and laboratory test results (total blood cell counts, coagulation factors, creatininemia, albuminemia, and liver function tests). The tumor response was determined by an assessment of the imaging studies by two radiologists. Follow-up imaging consisted of magnetic resonance imaging (MRI) in most patients and of computed tomography (CT) in a few. During imaging follow-up, 30 and 52 patients, 18 and 37 patients, 15 and 26 patients, 6 and 15 patients, 3 and 10 patients, and 2 and 2 patients underwent CT and MRI exams at 3, 6, 9, 12, 15, and 18 months, respectively. Follow-up imaging studies were compared to the imaging studies obtained at baseline (Figure 1A) and at other time points. The tumor response was evaluated using the Response Evaluation Criteria in Solid Tumors (RECIST) version 1.1 for mCRC and CCA, and the modified RECIST (mRECIST) for HCC to take into account the hypervascularized portion of the lesion at the arterial phase, considered as the active tumor, as recommended by guidelines [15,16].

For every follow-up imaging study, the disease response was classified as stable disease (SD), progressive disease (PD), partial response (PR), or complete response (CR). Progression-free survival (PFS) was defined as the time from the first treatment to hepatic progression on imaging studies according to mRECIST or RECIST. OS was defined as the time from first treatment to death. Data were censored on 31 December 2019.

### 2.4. Workup Procedure

The workup phase was carried out under local anesthesia. First, angiograms with selective hepatic artery catheterization and cone beam CT (CBCT) (Figure 1B,C) were performed to determine the hepatic vascularization pattern and to identify any non-target arteries that might lead to inadequate Y90 delivery, such as extrahepatic arterial branches arising from hepatic arteries and supplying extrahepatic organs. If a non-target artery was identified, it was embolized prophylactically using coils and/or glue. In some cases, CuraSpon^®^ (CuraMedical^®^, Assendelft, The Netherlands) was used to redirect the blood flow. CBCT images were carefully examined for target lesion coverage, relative lesion enhancement, and non-target perfusion. After defining the optimal placement of the microcatheter used for the treatment, 99 m Tc-MAA was injected. Then, 99 m Tc-MAA SPECT/CT was performed immediately to check the absence of a significant lung shunt (>20%) and of extrahepatic uptake. Liver and tumor volumes were determined on the last MRI or CT scan to calculate the optimal activity dose. Activity was calculated by nuclear physicists using the partition model if possible and the body surface area (BSA) method otherwise. The BSA method is semi-empirical and considers only the tumor and liver volumes. In contrast, the partition model is based on medical internal radiation dose principles and differentiates three discrete vascular compartments, namely, the lungs, tumor, and nontumoral liver parenchyma. The avidity of each compartment for albumin macroaggregates is taken into account. The BSA method was used essentially in patients with mCRC if the conditions for using the partition model were unfavorable (no significant tumor uptake on the workup scintigraphy, lesions smaller than 1 cm that raised contouring challenges).

### 2.5. Treatment Phase

TARE was performed under local anesthesia, generally 2 weeks after the workup procedure. The microcatheter was positioned exactly at the location determined during the workup and Y90 microspheres were then slowly injected. Great care was required during the microsphere injection to avoid reflux. Y90 SIR-Spheres^®^ (Sirtex Inc., Cosgrove, Australia) or TheraSpheres^®^ (MDS, Nordion, Ottawa, ON, Canada) were used. Y90 SIR-Spheres^®^ were always used if two or more doses were necessary, as well as in all patients with mCRC. Bilobar TARE procedures were performed in one or two sessions depending on liver function. A Photon-emission tomography (PET)-CT scan was performed on the following day to assess Y90 uptake (Figure 1D).

### 2.6. Follow-Up

Baseline laboratory tests (total blood cell counts, coagulation factors, creatininemia, albuminemia, and liver function tests) were done before TARE. Symptoms were evaluated during a 48 or 72 h hospitalization after TARE and the laboratory tests were repeated 24 h after TARE to evaluate toxicity and adverse events. Adverse events were reported using the Cardiovascular and Interventional Radiological Society of Europe (CIRSE) classification system [17]. Early side effects were defined as side effects appearing within 1 week after TARE. Ultrasonography and Doppler of the artery puncture site were performed on the day following TARE to look for vascular complications such as a false aneurysm. If no adverse event was recorded within 48 h following treatment, the patient was discharged. Follow-up visits were scheduled every 3 months for a clinical examination, laboratory tests, and imaging studies, to evaluate toxicity and the treatment response. Median follow-up was 32.1 months.

### 2.7. Statistical Tests

Values are described as mean ± SD (range) for normally distributed quantitative variables and as n (%) for qualitative variables. Chi^2^ and Shapiro-Wilk tests were used to analyze patient characteristics and the normality of data. Kaplan-Meier survival analyses were performed. Median survival with the corresponding 95% confidence intervals (95%CIs) were computed using the Kaplan Meier method. Kaplan-Meier survival curves were drawn for PFS in all patients, PFS separately in males and females, and OS. Univariate and multivariate analyses were conducted using Chi² tests and a Cox regression model, respectively, to identify population characteristics associated with survival. The statistical analysis was carried out using Stata software (StataCorp LLC, College Station, TX, USA). Statistical significance was established for *p* values less than 0.05.

## 3. Results

### 3.1. Patient Characteristics

During the study period, 70 consecutive patients underwent Y90 TARE at our institution. Table 1 lists their main characteristics. Of the 33 (47.1%) patients with cirrhosis, 84.8% were well compensated (Child Pugh A); among them, 32 had HCC and 1 had CCA. Of the 21 patients with portal thrombosis, 15 had HCC. The mean number of tumors was 3 in patients with HCC, 7 in patients with mCRC, and 1 in patients with CCA.

### 3.2. Treatments Given Before Y90 TARE

Table 2 reports the treatments given before Y90 TARE. Of the 34 (48.6%) patients given systemic chemotherapy, 10 had HCC, 4 had CCA, and 20 had mCRC. The mean number of lines was three. Sorafenib was used in most of the patients with HCC. Most of the 23 patients who received TACE had HCC; idarubicin and lipiodol were used.

### 3.3. Y90 TARE Treatment Characteristics

Patients were treated with TARE with four different intentions: to ensure survival while waiting for a liver transplant in 3 (4.3%) patients, for downstaging before surgical resection in 11 (15.7%) patients, as ablative therapy in 1 (1.4%) patient, and for palliative treatment in 55 (78.6%) patients. The mean number of Y90 administration sites was 1.4 (range, 1.0–3.0). Dose adjustments were required for 3 patients with lung shunting >10%. The mean and median doses delivered were 1.8 GBq and 1.6 GBq, respectively. Main data are summarized in Table 3.

### 3.4. Safety

After Y90 treatment, the hospital stay length for monitoring was 24–36 h in 6 (8.6%) patients, 36–48 h in 43 (61.4%) patients, 48–72 h in 13 (18.6%) patients, and longer than 72 h in 8 (11.4%) patients. There were no significant differences in hospital stay lengths according to type of malignancy (*p* = 0.6). Early side effects, mainly grade 1, were reported for 12 patients (Table 4). Only 2 patients experienced grade 3 side effects: 1 had radiation-induced cholecystitis due to nontargeted Y90 microsphere distribution and the other had angiocholitis. All these events were transient and responded to medical or radiological therapies. There were no cases of hepatic dysfunction or death related to TARE.

### 3.5. Efficacy and Tumor Response

Median follow-up was 964 days (32.1 months) (range, 769–1069). During follow-up, disease progression occurred in 63 (90.0%) patients. The disease remained controlled at the end of the study in 7 (10.0%) patients, as follows: complete response in 3 patients (HCC = 1, mCRC = 1, CCA = 1), partial response in 3 patients (HCC = 2, mCRC = 1), and stable disease in 1 patient (CCA = 1). All were alive. Median Kaplan-Meier PFS was 168 days (5.6 months) (range, 131–209) (Figure 2). PFS was significantly longer in women than in men (Hazard ratio, HR, 0.49; 95%CI, 0.26–0.90; *p* = 0.031) with a median of 452 days (15.1 months) (range, 133–823) compared to 166 days (5.5 months) (range, 97–200) for men (Figure 3). PFS was not significantly associated with age, primary malignancy, presence of cirrhosis, cause of cirrhosis, tumor distribution, number of tumors, or presence of portal thrombosis. Previous treatment by radiofrequency or microwave ablation was significantly associated with PFS, but only 8 patients had received this treatment. Previous treatment by TACE, hepatic arterial chemo-infusion, or systemic chemotherapy was not significantly associated with PFS (*p* = 0.6, *p* = 0.9, and *p* = 0.2, respectively). The risk of progression increased with the number of chemotherapy lines (HR, 1.2; 95%CI, 0.9–1.3, *p* = 0.05). The risk of progression increased by 16.0% for each additional chemotherapy line according to the Cox model. Given the small number of patients, no test was done assessing the survival difference between non-responders and responders.

### 3.6. Overall Survival

During follow-up, 47 (67.1%) patients died after a median of 964 days (32.1 months) (range, 769–1069 days). The median OS was 16.1 months (range, 1.0–72.5 months) from Y90 TARE to death or study completion in November 2018 (Figure 4). There were no significant differences in survival for patients with HCC (*p* = 0.5), mCRC (*p* = 0.2), or CCA (*p* = 0.09) (Table 5). Men were at significantly higher risk of death than women (HR, 0.3; 95%CI, 0.1–0.7; *p* = 0.009, Kaplan Meier method). In contrast, the risk of death was not significantly associated with age; tumor distribution; presence of portal thrombosis; cirrhosis, cause of cirrhosis, or Pugh-Child score; or receiving TACE before TARE.

The risk of death increased with the number of chemotherapy lines (HR, 1.2; 95%CI, 1.0–1.5, *p* = 0.026) according to the Cox model. The risk of death increased by 23.0% for each additional chemotherapy line. Overall, 8 patients continued to take systemic therapy in conjunction with TARE: 7 HCC patients, 1 mCRC patient, and 1 CCA patient.

## 4. Discussion

In our retrospective study, TARE was safe and efficient for treating intermediate or advanced stage HCC, CCA, and mCRC with tumor refractoriness or patient intolerance to appropriate treatments. The 17% frequency of side effects, with no significant difference according to tumor type, is acceptable in the light of the current literature [18]. The median PFS and median OS were 168 days (range, 131–209) and 16.1 months (range, 1.0–72.5), respectively.

In our study, the overall median OS was 16.2 months with no significant differences according to tumor type. This result is consistent with previous publications [19,20,21,22]. The median OS for patients with HCC was 18.1 months (range, 14.4–39.0). In previous studies, the median OS after TARE in BCLC B and C patients ranged from 10 to 16.9 months without prior TACE [19,20] and was 8.4 months with prior TACE [20,21]. Many HCC prognostic scores are available for survival analysis [23]. It is important to remember that TARE is mainly used as a palliative treatment. As a reminder, when curative treatments are used for HCC patients, the 5-year survival rate after surgical resection for patients without portal hypertension and normal bilirubin levels is around 75% but significantly decreases in case of portal hypertension [24]. Liver transplantation in patients meeting Milan criteria allows a 70% survival rate in 5 years. Lastly, it has been shown that ablative therapies are able to achieve complete tumor removal in more than 90% of HCC cases with a single tumor of less than 4 cm [24]. In our population, prior TACE was not significantly associated with OS [22,25]. Median OS for patients with mCRC in our study was 12.9 months (range, 5.1–28.4). Studies of patients with liver colorectal cancer metastases at different disease stages, with or without chemotherapy, found that median OS after TARE ranged from 10.5 to 29.4 months [26,27,28,29]. In the two randomized controlled trials, survival was significantly better in the TARE arm [26,27]. The OS of patients with CCA has been estimated at 8.1 to 11.7 months when treated only with systemic chemotherapy [30,31]. In our study, TARE seemed to improve survival compared to chemotherapy only, with a median OS of 16.8 months (range, 3.7–NA) [32,33]. However, only 6 patients had CCA in our study. Several ongoing prospective studies have produced encouraging intermediate results and may confirm the role for TARE in the therapeutic arsenal for CCA [34,35]. Indeed, the first results from the CIRSE registry in Europe using SIR-Spheres therapy recently reported and analyzed 1027 patients in real-life conditions [36]. Overall, 68.2% of the intention of treatment was palliative. Up to half of the patients received systemic therapy and/or locoregional treatments prior to TARE (53.1% and 38.3%, respectively). Median OS was reported per cohort and was 16.5 months for HCC, 14.6 months for CCA. For mCRC patients, median OS was 9.8 months [36]. Better timeline management may improve OS. The times from baseline imaging and from the multidisciplinary committee decision to TARE (mean, 67.1 days and 56.6 days, respectively) were long in our study and may partially explain the treatment failures seen in these patients at high risk for disease progression. The long-time delays may be explained by referral of some patients by secondary general hospitals which can be time-consuming.

TARE appeared safe, with a low rate of generally mild side effects, in keeping with previous reports. One study found post-embolization syndrome in 50% of patients, as well as grade 3 side effects such as radioembolization-induced liver disease and radiation-induced gastric ulceration or cholecystitis, in 1% to 5% of patients [37]. The workup phase aimed at avoiding the irradiation of non-targeted tissues, careful patient selection, the use of dosimetry models, and technical expertise help to reduce the incidence of complications [38,39].

The risk of death and disease progression increased with the number of previous chemotherapy lines in our study. Malignancies requiring numerous chemotherapy lines are likely to be more aggressive. In addition, chronic exposure to chemotherapy, such as bevacizumab, may induce changes in tumor vascularization that may decrease the response rate to Y90 microspheres [40]. We found a significantly higher risk of death and progression in men compared to women. However, the usually greater burden of comorbidities and shorter life expectancy in men may have acted as confounding factors. It has also been hypothesized that estrogens may protect against cancer progression, notably for colorectal cancer [41,42].

Our study has several limitations. First, we used a single-center retrospective design. Second, the sample size was small, with only 6 CCA patients. The population is heterogenous, with three different malignancies treated at different stages and with or without prior treatments. Third, two different Y90 microspheres were used. Resin (SIR-Spheres^®^) and glass (TheraSpheres^®^) microspheres differ in density, specific activity, and particle counts and may therefore produce different results. Last, data concerning the estimated absorbed doses to the liver and to the tumor were not available for this study. It might have an impact on the response tumor and survival. However, our study reports real-life experience with Y90 TARE in a tertiary academic French center, and results are in accordance with those of previous registries [36,43].

## 5. Conclusions

In conclusion, our study confirms that TARE can be a safe and effective treatment option for patients with HCC (according to Barcelona Clinic Liver Cancer Classification recommendations), mCRC, or CCA who relapse after appropriate treatment (surgical, ablative, vascular, or systemic).

## Figures and Tables

**Figure 1 diagnostics-11-00122-f001:**
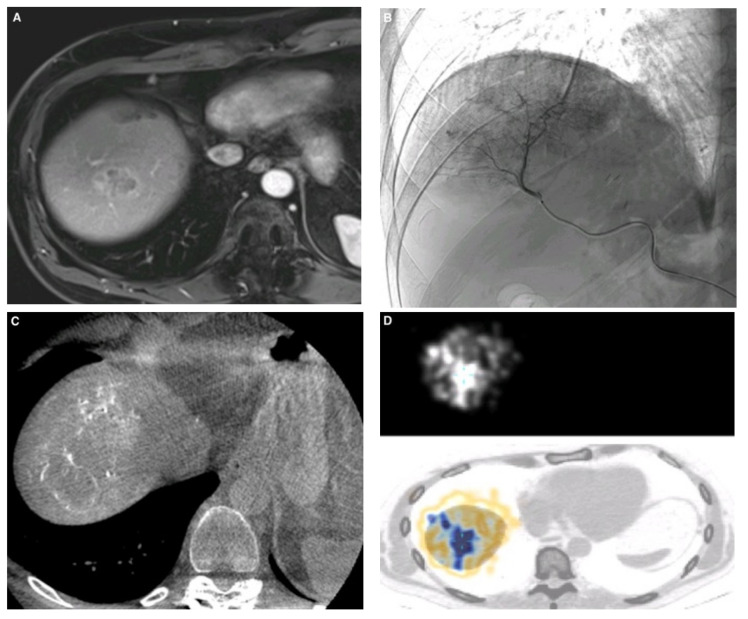
51-year-old man with a single liver metastasis from colorectal cancer. (**A**) Hepatic magnetic resonance imaging (MRI) before Y90 transarterial radioembolization (TARE) showing contrast enhancement of the metastasis in the hepatic dome. (**B**) Selective angiography of arteries supplying the VII and VIII segments of the liver, obtained with a Progreat 2.4 microcatheter before the injection of Y90 microparticles. (**C**) Cone beam CT with contrast injection showing good targeted-lesion coverage with the microcatheter in the appropriate position. (**D**) Photon-emission tomography (PET) CT scan performed on the day following TARE, showing uptake by the targeted liver tumor.

**Figure 2 diagnostics-11-00122-f002:**
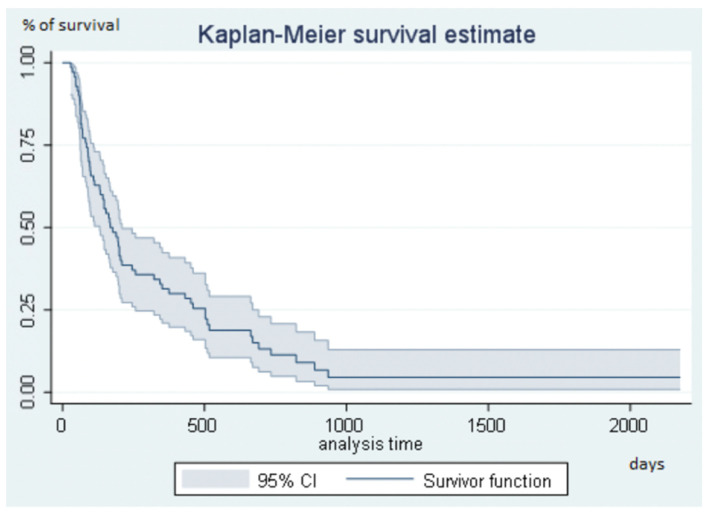
Progression-free survival in the overall study population.

**Figure 3 diagnostics-11-00122-f003:**
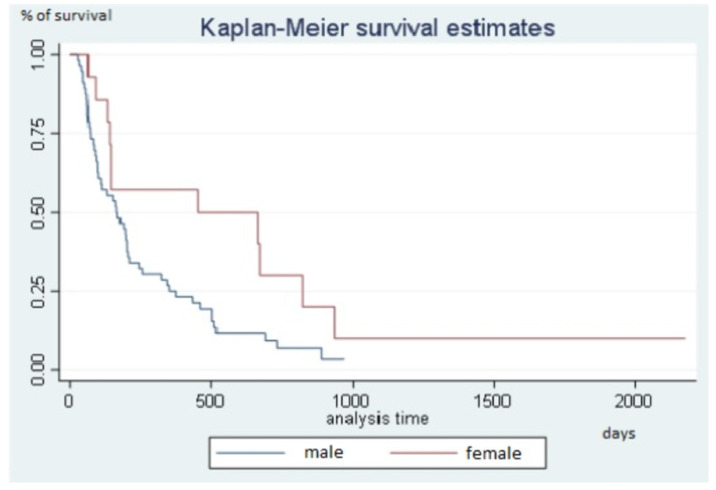
Progression-free survival in males and females.

**Figure 4 diagnostics-11-00122-f004:**
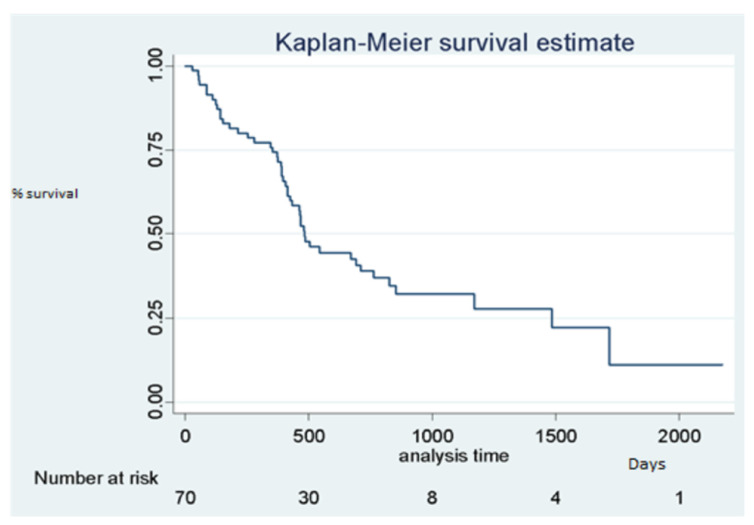
Overall survival. Kaplan-Meier analysis showing a median overall survival of 482 days (16.1 months) (range, 1–72.5 months) after Y90 TARE.

**Table 1 diagnostics-11-00122-t001:** Patients’ characteristics.

Characteristics	No. of Patients (%)
Age (years)	
Mean ± SD	62.5 ± 10.1
Median (range)	63 (33–86)
Sex	
Male	56 (80.0)
Female	14 (20.0)
ECOG score	
0	45 (64.3)
1	25 (35.7)
Underlying malignancy	
HCC / BCLC A/BCLC B/BCLC C	44 (62.9) / 0 (0)/29 (65.9)/15 (34.1)
mCRC	20 (28.6)
CCA	6 (8.5)
Tumor characteristics	
Extrahepatic disease	
Yes	4 (5.7)
No	66 (94.3)
Liver disease distribution	
Unilobar	43 (61.4)
Bilobar	22 (31.4)
Segmental	5 (7.2)
Portal thrombosis	
Yes	21 (30.0)
Major	3 (4.3)
Segmental	18 (25.7)
No	49 (70.0)
No. of liver lesions	
1	26 (37.1)
2	9 (12.9)
3	6 (8.6)
4	5 (7.1)
5 to 10	11 (15.7)
>10	13 (18.6)
Liver cirrhosis	
Yes	33 (47.1)
No	37 (52.9)
Cause of liver cirrhosis	
Alcohol abuse	22 (66.7)
NASH	4 (12.1)
Hepatitis B	1 (3.0)
Hepatitis C	5 (15.2)
Hemochomatosis	1 (3.0)
Child-Pugh score	
A5	17 (51.5)
A6	12 (36.4)
B7	3 (9.1)

ECOG score, Eastern Cooperative Oncology Group Performance Status score; HCC, hepatocellular carcinoma; BCLC, Barcelona Clinic Liver Cancer; mCRC, colorectal cancer hepatic metastases; CCA, cholangiocarcinoma; NASH, nonalcoholic steatohepatitis.

**Table 2 diagnostics-11-00122-t002:** Treatment before transarterial radioembolization (TARE).

Treatment	No. of Patients (%)
Surgical resection	19 (27.1)
Liver transplantation	1 (1.5)
Ablative procedure	8 (11.4)
RFA	6 (8.6)
MWA	2 (2.9)
Intra-arterial treatment	31 (44.3)
HAI	8 (11.4)
TACE	23 (32.9)
Systemic chemotherapy	34 (48.6)
No. of lines	
1	14 (20.0)
2	9 (12.9)
3	4 (5.7)
4	4 (5.7)
5	1 (1.4)
6	2 (2.9)

RFA, radiofrequency ablation; MWA, microwave ablation; HAI, hepatic arterial infusion; TACE, transarterial chemoembolization.

**Table 3 diagnostics-11-00122-t003:** Characteristics of transarterial radioembolization (TARE).

Characteristics	No. of Patients (%) or Mean ± SD (Range)
Y90 microspheres	
SIR-Spheres^®^ *	41 (58.6)
TheraSphere^®^ **	29 (41.4)
Prophylactic extrahepatic branch occlusion	
No	47 (67.1)
Yes	23 (32.9)
Gastro-duodenal artery	5 (21.7)
Right gastric artery	4 (17.4)
Left gastric artery	2 (8.7)
Other arteries	12 (52.2)
Time from baseline imaging to TARE, days	67.1 ± 38.9 (10–182)
Time from multidisciplinary committee decision to TARE, days	56.6 ± 23.6 (13–133)
Time from work-up to TARE, days	14.4 ± 5.0 (3–42)
Lung shunt, %	3.9 ± 3.1
Activity delivered, GBq	1.8 ± 1.2
Time from baseline imaging to TARE, days	67.1 ± 38.9 (10–182)

* Sirtex Inc., Cosgrove, Australia; ** MDS Nordion, Ottawa, Ontario, Canada.

**Table 4 diagnostics-11-00122-t004:** Clinical toxicities of transarterial radioembolization (TARE).

Toxicity	No. of Patients (%)
Nausea	3 (4.2)
Abdominal pain	7 (10.0)
Fever	1 (1.4)
Vascular-like pseudoaneurysm	2 (2.8)
Radiation-induced cholecystitis	1 (1.4)
Radiation-induced hepatitis	0 (0.0)
Angiocholitis	1 (1.4)

**Table 5 diagnostics-11-00122-t005:** Kaplan-Meier estimates of overall survival (OS).

Variable	No. of Patients (%)	Median OS (95%CI)
Malignancy		
HCC	44 (62.9)	18.1 (14.4–39.0)
mCRC	20 (28.6)	12.9 (5.1–28.4)
CCA	6 (8.5)	16.8 (3.7–NA)
Sex		
Male	56 (80.0)	15.5 (13.0–18.1)
Female	14 (20.0)	NA

OS, overall survival; HCC, hepatocellular carcinoma; mCRC, colorectal cancer hepatic metastases; CCA, cholangiocarcinoma; NA, not applicable.

## Data Availability

The data presented in this study are available on request from the corresponding author. The data are not publicly available due to identity reasons.

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
