# Peer review of "Transarterial Radioembolization of Hepatocellular Carcinoma, Liver-Dominant Hepatic Colorectal Cancer Metastases, and Cholangiocarcinoma Using Yttrium90 Microspheres: Eight-Year Single-Center Real-Life Experience"

_diagnostics, 2021, doi:10.3390/diagnostics11010122_

Round 1

Reviewer 1 Report

GENERAL COMMENT: Well written retrospective evaluation of TARE in a variety of hepatic malignancies.

INTRODUCTION: Authors are advised to comment (and include in the reference list) the CIRT registry paper (Helmberger, T., Golfieri, R., Pech, M. et al. Clinical Application of Trans-Arterial Radioembolization in Hepatic Malignancies in Europe: First Results from the Prospective Multicentre Observational Study CIRSE Registry for SIR-Spheres Therapy (CIRT). Cardiovasc Intervent Radiol 44, 21–35 (2021). https://doi.org/10.1007/s00270-020-02642-y)

MM: Authors are advised to provide more details upon the study flow. Were the patients included in the study consecutive? Are these the only patients with hepatic malignancies treated with TARE in the mentioned time frame? Were any patients excluded from the study (if yes why?).

MM: Authors are advised to provide absolute numbers of patients undergoing MRI or CT during follow-up imaging

MM: Authors need to provide a comment/explanation on why the y have used mRECIST for HCC and RECIST criteria for cholangio and colorectal mets

MM: please provide vendor details for Gelitaspon

Authors are kindly asked to report complications and adverse events using the CIRSE classification system for complications reporting (Cirse Quality Assurance Document and Standards for Classification of Complications: The Cirse Classification System.Filippiadis DK, Binkert C, Pellerin O, Hoffmann RT, Krajina A, Pereira PL. Cardiovasc Intervent Radiol. 2017 Aug;40(8):1141-1146.)

Author Response

  1. English language and style are fine/minor spell check required.

Reply: Thanks for this comment. English language has been edited by a native speaker.

  1. GENERAL COMMENT: Well written retrospective evaluation of TARE in a variety of hepatic malignancies.

Reply:Thanks for this comment.

  1. INTRODUCTION: Authors are advised to comment (and include in the reference list) the CIRT registry paper (Helmberger, T., Golfieri, R., Pech, M. et al.Clinical Application of Trans-Arterial Radioembolization in Hepatic Malignancies in Europe: First Results from the Prospective Multicentre Observational Study CIRSE Registry for SIR-Spheres Therapy (CIRT). Cardiovasc Intervent Radiol44, 21–35 (2021).https://doi.org/10.1007/s00270-020-02642-y).

Reply:Thanks for this comment. This reference has been added in the list and commented in the discussion section, as suggested.

  1. MM: Authors are advised to provide more details upon the study flow. Were the patients included in the study consecutive? Are these the only patients with hepatic malignancies treated with TARE in the mentioned time frame? Were any patients excluded from the study (if yes why?).

Reply:Thanks for this comment. All patients included in the study were consecutive. Yes, these are the only patients with hepatic malignancies treated with TARE in the mentioned time frame. Only patients with no favourable work-up were excluded from the study because not treated (already specified in the exclusion criteria in the study design section).  It has been specified in the 2.2 and 3.1 paragraph of the materials and methods and results sections.

  1. MM: Authors are advised to provide absolute numbers of patients undergoing MRI or CT during follow-up imaging.

Reply:Thanks for this comment. Numbers have been provided in the text in the data collection section paragraph of the materials and methods section for both MRI and CT exams during follow-up, as suggested.

  1. MM: Authors need to provide a comment/explanation on why they have used mRECIST for HCC and RECIST criteria for cholangio and colorectal mets.

Reply: Thanks for this comment. The evaluation of tumor response in oncology is mainly based on RECIST 1.1. It was used as part of guidelines for CCA and mCRC. However, it is well known that these criteria are not sufficient for the therapeutic evaluation of HCC of which main current treatments consist in destroying the lesion resulting in non-viable tissue. The guidelines recommend to use mRECIST (modified RECIST) criteria for HCC specifically. They were implemented in order to take into account the hypervascularized portion of the lesion at the arterial phase, considered as the active tumor, for the follow-up. It has been specified in the text in the data collection paragraph of the materials and methods section.

  1. MM: please provide vendor details for Gelitaspon.

Reply:Thanks for this comment. Vendor details have been provided in the text. CuraSponÒ(CuraMedicalÒ, Assendelft, The Netherlands) was used.

  1. Authors are kindly asked to report complications and adverse events using the CIRSE classification system for complications reporting (Cirse Quality Assurance Document and Standards for Classification of Complications: The Cirse Classification System. Filippiadis DK, Binkert C, Pellerin O, Hoffmann RT, Krajina A, Pereira PL.Cardiovasc Intervent Radiol. 2017 Aug;40(8):1141-1146.).

Reply:Thanks for this comment. Complications and adverse events have been reported using this classification. The article has been added to the reference list.

Reviewer 2 Report

Nice study.

Can you provide the following?

What was the BCLC spread in your patients? How many A, B,C? Did that have any survival impact?

What was the main type or response to Y90, Stable disease? Complete response? or PR? Was there a survival difference between non responders and responders

Very strange that there were no cases of liver dysfunction following Y90. What was the estimated absorbed dose to the liver? What was the dose to the tumor?

How many patients continued to take systemic therapy in conjunciton with Y90?

Author Response

  1. English language and style are fine/minor spell check required.

Reply: Thanks for this comment. English language has been edited by a native speaker.

  1. Nice study.

Reply:Thanks for this comment.

  1. Can you provide the following?

Reply:Thanks for this comment. It has been provided below.

  1. What was the BCLC spread in your patients? How many A, B, C? Did that have any survival impact?

Reply:Thanks for this comment. The BCLC was as follows: A (0 patients), B (29 patients), C (15 patients). It has been added in Table 1. No statistical test was specifically done regarding the impact of BCLC grade on survival but each criteria (thrombosis, ECOG, cirrhotic stade…) were tested instead. It’s already mentioned in the 3.5 paragraph of the results section.

  1. What was the main type or response to Y90, Stable disease? Complete response? or PR?

Reply:Thanks for this comment. The type of response at the end of the study was as follows: complete response (HCC=1, mCRC=1, CCA=1), partial response (HCC=2, mCRC=1), stable disease (CCA=1). It has been added in the text in the 3.5 paragraph of the results section.

  1. Was there a survival difference between non-responders and responders?

Reply:Thanks for this comment. All 7 patients with disease which remained controlled at the end of the study were alive. Given the small number of patients and lack of power, no statistical test was done to evaluate survival difference between non-responders and responders. It has been added in the 3.5 paragraph of the results section.

  1. Very strange that there were no cases of liver dysfunction following Y90. What was the estimated absorbed dose to the liver? What was the dose to the tumor?

Reply:Thanks for this comment. As mentioned, no REILD was reported. Hopefully it is a rare complication. Mean activity delivered was 1.8±1.2 GBq. Estimated absorbed doses to the liver and to the tumor were not specifically evaluated for this study and are not available. It has been cited as one more limitation of the study in the limitations section.

  1. How many patients continued to take systemic therapy in conjunction with Y90?

Reply:Thanks for this comment. Overall, 8 patients continued to take systemic therapy in conjunction with TARE, 7 HCC patients, 1 mCRC patient and 1 CCA patient. It has been added in the 3.6 paragraph of the results section.

Reviewer 3 Report

The study is very interesting especially considering the increasing incidence of HCC in the general population and that patients with this kind of tumor are not always candidate to surgical interventions.

That being considered, I suggest to add in the statistical section if any statistical test was used to assess the normality of data.

Also, I deeply suggest to cite this recent paper in your discussion: DOI: 10.4254/wjh.v12.i12.1239

In addition, in the discussion section the authors should stress the difference in survival between TARE vs surgical and other "curative" treatments for HCC.

Author Response

1. The study is very interesting especially considering the increasing incidence of HCC in the general population and that patients with this kind of tumor are not always candidate to surgical interventions.

Reply:Thanks for this comment. We fully agree.

2. That being considered, I suggest to add in the statistical section if any statistical test was used to assess the normality of data.

Reply:Thanks for this comment.Yes, Chi2and Shapiro-Wilk tests were used to analyse patient characteristics and the normality of data. It has been added in the statistical section. Our statistician has been added among the authors and author contribution changed consequently at the end of the manuscript.

3. Also, I deeply suggest to cite this recent paper in your discussion: DOI:10.4254/wjh.v12.i12.1239.

Reply:Thanks for this comment. This reference has been cited in the discussion section and added to the reference list as suggested.

4. In addition, in the discussion section the authors should stress the difference in survival between TARE vs surgical and other "curative" treatments for HCC.

Reply:Thanks for this comment. The difference in survival and response between TARE and other curative treatments has been discussed in the discussion section as suggested. One more reference below has been added and references have been renumbered (Kumari, R.; Sahu, M.K.; Tripathy, A.; Uthansingh, K.; Behera, M. Hepatocellular carcinoma treatment: hurdles, advances and prospects. Hepat Oncol 2018, 5, HEP08,doi: 10.2217/hep-2018-0002).